# Exploring ligand binding pathways on proteins using hypersound-accelerated molecular dynamics

Mitsugu Araki [1✉], Shigeyuki Matsumoto [2], Gert-Jan Bekker [3], Yuta Isaka[4], Yukari Sagae[1], Narutoshi Kamiya[5] & Yasushi Okuno [1,2✉]

Capturing the dynamic processes of biomolecular systems in atomistic detail remains difficult despite recent experimental advances. Although molecular dynamics (MD) techniques enable atomic-level observations, simulations of "slow" biomolecular processes (with timescales longer than submilliseconds) are challenging because of current computer speed limitations. Therefore, we developed a method to accelerate MD simulations by high-frequency ultrasound perturbation. The binding events between the protein CDK2 and its small-molecule inhibitors were nearly undetectable in 100-ns conventional MD, but the method successfully accelerated their slow binding rates by up to 10–20 times. Hypersound-accelerated MD simulations revealed a variety of microscopic kinetic features of the inhibitors on the protein surface, such as the existence of different binding pathways to the active site. Moreover, the simulations allowed the estimation of the corresponding kinetic parameters and exploring other druggable pockets. This method can thus provide deeper insight into the microscopic interactions controlling biomolecular processes.

[1] Graduate School of Medicine, Kyoto University, Kyoto, Japan. [2] Medical Sciences Innovation Hub Program, RIKEN Cluster for Science, Technology and Innovation Hub, Yokohama, Japan. [3] Institute for Protein Research, Osaka University, Osaka, Japan. [4] Research and Development Group for In Silico Drug Discovery, Center for Cluster Development and Coordination (CCD), Foundation for Biomedical Research and Innovation at Kobe (FBRI), Kobe, Japan. [5] Graduate School of Simulation Studies, University of Hyogo, Kobe, Japan. ✉email: araki.mitsugu.6w@kyoto-u.ac.jp; okuno.yasushi.4c@kyoto-u.ac.jp

The microscopic observation of biomolecular processes such as protein folding, protein interactions, and enzyme reactions, most of which occur on timescales ranging from microseconds to seconds[1], is of great interest to the molecular biology community. Although molecular dynamics (MD) simulations enable atomic-level observations, they are limited to several microseconds on standard high-performance computers and are thus normally applicable only to relatively fast processes[2]. Recently, the kinetics of slower protein interaction processes were explored through long MD simulations spanning timescales of tens of microseconds to milliseconds[3–6], which were achieved through the development of special-purpose supercomputers for high-speed simulations (e.g., ANTON[7]) and/or algorithms to aggregate many short simulations (e.g., Markov state models (MSMs)[8]). Unfortunately, MD-specific supercomputers such as ANTON are accessible only to a limited number of researchers owing to their limited computational resources. While MSMs have a lower requirement for simulation power, this method is very sensitive to the choice of hyperparameters[9], which makes MSM approaches less than straightforward to use.

To overcome these problems, we have developed an MD simulation method that utilizes high-frequency ultrasound (hereafter denoted as hypersound) shock waves to accelerate the dynamics. This method falls into the category of nonequilibrium MD simulations under external field perturbation[10,11]. Its key advantage is that it allows naturally "slow" processes such as those mentioned above to be frequently and directly captured in a series of single MD trajectories performed on standard high-performance computers. In the experimental field, ultrasound irradiation procedures have been applied to accelerate various kinds of chemical reactions[12,13] and to synthesize nanoparticles[14]. This acceleration is considered to be induced by acoustic cavitation (i.e., the repeated growth and collapse of cavitation bubbles formed by the ultrasound waves, which generate local high-temperature/pressure regions in solution)[15]. Inspired by these results, in this study, we first analyzed the hypersound-dependent behavior of a liquid water model to test the effect of shock waves with a protein-size wavelength. Next, to assess the effect of hypersound acceleration on biomolecular processes, we performed short (100–200 ns) simulations to capture the slow binding of small-molecule inhibitor compounds (CS3 and CS242) to cyclin-dependent kinase 2 (CDK2)[16], as a representative system in which the binding event would be nearly undetectable in standard MD. The simulations showed a significant acceleration of the binding process under hypersound irradiation compared to standard MD simulations. The hypersound-accelerated simulations revealed the existence of various conformationally and energetically diverse binding pathways, suggesting that the assumption of a single pathway/transition state made in conventional kinetic models may be inaccurate. The present method allowed not only the estimation of kinetic parameters of slow-binding inhibitors but also the full exploration of druggable sites. This approach would thus be helpful for efficiently understanding the microscopic mechanism of slow biomolecular processes.

## Results

**Hypersound-perturbed MD simulation of liquid water.** To simulate hypersound shock waves with protein-size wavelengths, their frequency was set to 625 GHz (corresponding to a period of 1.6 ps) (Fig. 1A), which is more than 100 times higher than that of currently used ultrasound waves[17]. The hypersound-perturbed MD simulation of liquid water at 298 K showed the generation and propagation of a high-density region (Fig. 1B and Supplementary Movie 1). Then, we analyzed the MD trajectory focusing on wave propagation along the $X$ direction as a representative

example (Fig. 1C–F). As the $X$ coordinate of the first wave reached 4 nm at a simulation time of 1.7 ps after passing through $X = 2$ nm at 0.7 ps (Fig. 1E), the propagation speed of the shock wave could be estimated to be 2000 m/s, which is similar to the speed of sound in water (~1500 m s$^{-1}$)[18]. The wavelength of the simulated shock wave was estimated to be 3.2 nm (2000 m s$^{-1}$ × 1.6 ps), which corresponds to the hydrodynamic radii of globular proteins consisting of 300–400 residues[19]. This confirms the successful generation of hypersound waves, which is appropriate to perturb biomolecular processes. The pressure ($p_x$) and kinetic energy ($k_x$) of the liquid water model also exhibited periodic fluctuations with the same phase as the density, reaching ~2000 atmospheres and 0.4–0.5 kcal/mol (corresponding to 400–500 K) at the center ($X = 4$ nm) of the simulation box (Fig. 1C, D, F). In contrast, the macroscopic properties of liquid water were not affected by the hypersound irradiation, except for the diffusion constant, which slightly increased to $6.30 \pm 0.10 \times 10^{-5}$ cm$^2$ s$^{-1}$, equivalent to the corresponding parameter of bulk water at 305 K (Supplementary Table 1). These results demonstrate that hypersound irradiation of a liquid solvent generates local higher pressure and temperature regions appropriate for promoting chemical processes[15] without altering the macroscopic properties of the liquid.

**CDK2–ligand-binding simulation under hypersound irradiation.** We next assessed the influence of shock wave perturbation on the association between the CDK2 protein and its slow-binding ATP-competitive inhibitors CS3 and CS242. The probability of observing the ligand-binding event in conventional MD simulations of 100 ns was estimated to be only 0.7% (2/283, corresponding to 2 out of 283 MD runs resulting in binding) for CS3 and 0.5% (2/369) for CS242 (Supplementary Table 2). On the other hand, higher probabilities were attained in 100-ns long hypersound-perturbed MD simulations. Hypersound irradiation with a higher amplitude or frequency resulted in an increase in the probability of observing ligand binding without significant loss of computation speed (Supplementary Table 3). When using a parameter set of ($N = 50$ and $v_{max} = 400$ m/s), the CS3 and CS242 binding probabilities were 12.4% (22/177) and 4.8% (11/227), respectively (Supplementary Table 2), showing that the perturbation successfully increased the association rate by 17.7 times (12.4/0.7) for CS3 and 9.6 times (4.8/0.5) for CS242. MD trajectories that exhibited stable ligand binding were extended to 200 ns to observe the behavior of the bound ligand, and based on their percentages, the association rate constants ($k_{on}$) under hypersound irradiation, using the same parameters as above, were estimated to be $3.68 \times 10^6$ for CS3 and $1.92 \times 10^6$ M$^{-1}$ s$^{-1}$ for CS242 (Table 1). This analysis proved the effectiveness of the hypersound-perturbed simulations in enhancing the sampling of infrequent binding events; this approach can thus be applied to extract further atomic-level information on these processes, as follows.

**Conformationally and energetically diverse binding pathways.** Hypersound-accelerated MD simulations revealed that multiple transitions between different conformations took place within each individual binding pathway (see Fig. 2A and Supplementary Movie 2 for CS3 and Fig. 2B and Supplementary Movie 3 for CS242). This emerges from the inspection of the 67 (CS3) and 14 (CS242) binding pathways observed in the hypersound-perturbed MD simulations, a few representative cases of which are shown in Supplementary Figs. 1 and 2. It should be noted that these pathways contain those observed in conventional MD simulations (Supplementary Fig. 3). The potential energy trajectories (also displayed in the figures) reveal the occurrence of multiple energy

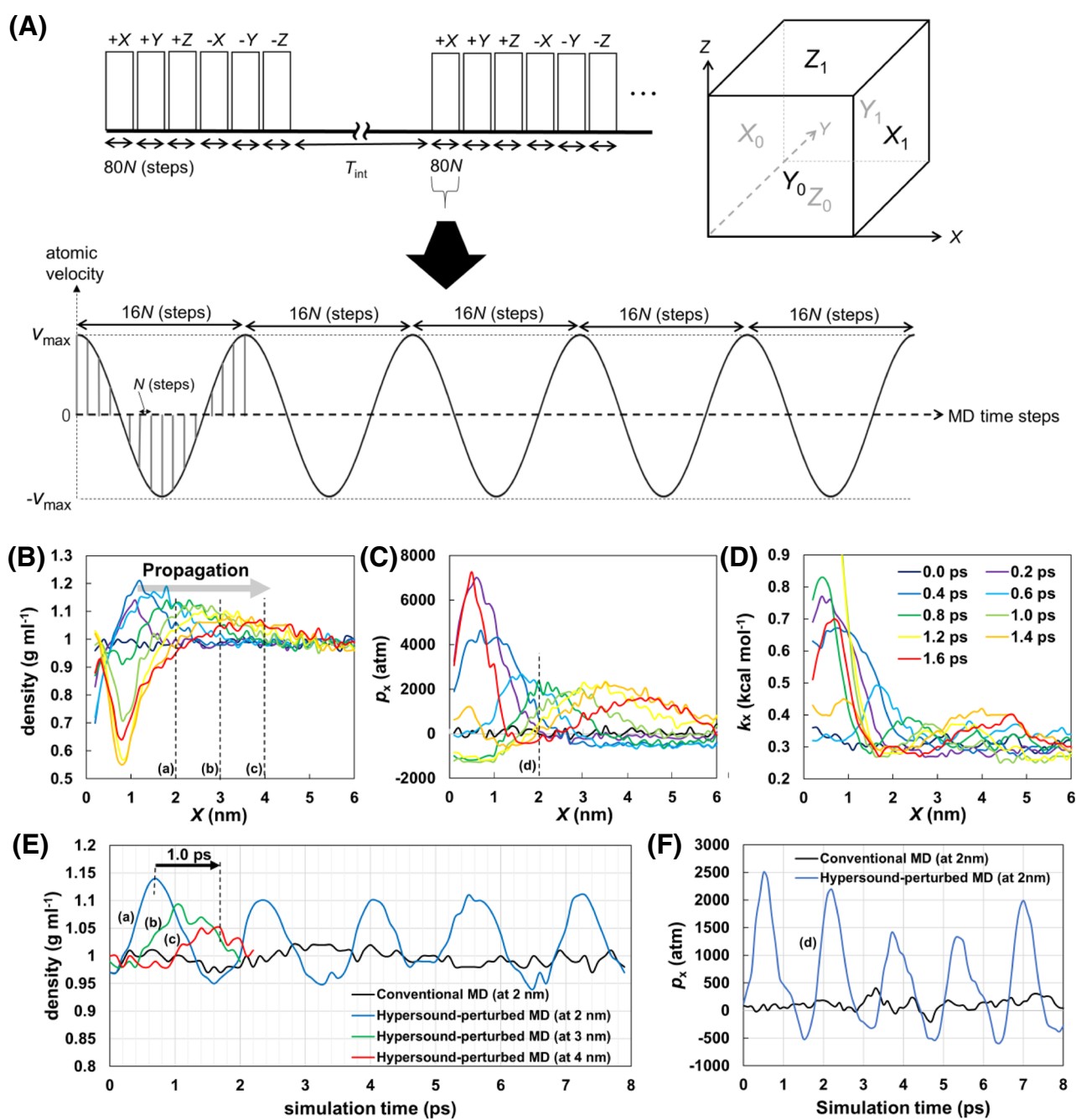

**Fig. 1 Schematic illustration of the modeling of hypersound shock waves in MD simulations and hypersound-perturbed water dynamics at 298 K.**
**A** (Top) Generation of hypersound shock waves in six directions ($+X$, $+Y$, $+Z$, $-X$, $-Y$, and $-Z$), propagating from the $X_0$, $Y_0$, $Z_0$, $X_1$, $Y_1$, and $Z_1$ surfaces, respectively, of the simulation box. (Bottom) Each shock wave consisted of 5 cycles and involved 80 ($=16 \times 5$ cycles) velocity pulses (indicated by vertical bars) applied every $N$ MD time steps (see "Methods" for details). **B**–**D** Spatial variation of **B** mass density, **C** pressure in the $+X$ direction ($p_x$), and **D** $X$ component of kinetic energy ($k_x$), measured at different simulation times. **E**, **F** Time dependence of **E** mass density and **F** the pressure, measured at different $X$ positions; the corresponding positions are shown in (**B**) and (**C**). Shock waves were generated in the $X = 0$–1 nm region [$X_0$ surface in (**A**)] of the simulation box.

barriers along each binding pathway and show that the position and height of the highest-energy transition state depend on the binding pathway (Fig. 2C). The trajectories indicate that the ligand tends to adopt energetically unstable configurations upon (i) entry into the CDK2 pocket (Fig. 2A, and Supplementary Figs. 1A and 2A) or (ii) conformational rearrangement in the pocket interior (Fig. 2B, and Supplementary Figs. 1B and 2B). These effects have not been previously captured by ensemble-averaged kinetic experiments[16,20] or existing generalized-ensemble MD simulations (Supplementary Fig. 3)[21], which predict a plausible pathway by efficiently exploring the conformational space. Ligand unbinding was also observed in some of these trajectories, most of which also exhibited different binding and unbinding pathways (Supplementary Figs. 1C and 2C). This suggests that the conventional kinetic model based on identical binding/unbinding pathways is not always valid at the single-

**Table 1 Kinetic parameters of the CDK2–ligand-binding process determined by conventional and hypersound-perturbed MD simulations. The values are presented as the mean ± SD.**

| | Association rate constant $k_{on}$ ($M^{-1} s^{-1}$) | Activation energy $E$ (kcal mol$^{-1}$)[a] | Diffusion constant $D$ (×10$^{-5}$ cm$^2$ s$^{-1}$) | Steric factor log ($\rho$) | Frequency factor log ($A$ ($M^{-1} s^{-1}$)) | Effective temperature $T$ (K) |
|---|---|---|---|---|---|---|
| CS3 (conventional MD) | $3.35 \times 10^{5}$[b] | 3.9 ± 1.8 | 0.17 ± 0.05 | −0.76 | 8.35 ± 0.13 | 298 |
| CS3 (hypersound-perturbed MD)[c] | $3.68 \times 10^{6}$ | 3.9 ± 1.8 | 0.61 ± 0.16 | −0.76 | 8.91 ± 0.12 | 362 |
| CS242 (conventional MD) | $3.21 \times 10^{4}$[b] | 6.7 ± 2.4 | 0.19 ± 0.07 | 0.20 | 9.36 ± 0.17 | 298 |
| CS242 (hypersound-perturbed MD)[c] | $1.92 \times 10^{6}$ | 6.7 ± 2.4 | 0.30 ± 0.10 | 0.20 | 9.56 ± 0.15 | 445 |

[a]The $E$ parameter was estimated from hypersound-perturbed MD simulations with $N = 50$ steps, $v_{max} = 400$ m/s, and $T_{int} = 2400$ N, assuming that the activation energy is not affected by the hypersound irradiation.
[b]Experimentally determined $k_{on}$ values retrieved from the Community Structure-Activity Resource (CSAR) database[16].
[c]Hypersound shock waves with the same parameters as above were used to determine the $k_{on}$, $D$, $A$, and $T$ parameters, which vary depending on the hypersound parameters (Supplementary Table 3).

molecule level. The trajectories of individual ligand molecules captured by the hypersound perturbation approach revealed the complex microscopic nature of the CDK2-inhibitor binding kinetics, highlighting the effectiveness of this approach in exposing effects not accessible by other experimental and computational techniques.

**Estimation of kinetic parameters of CDK2-ligand binding.** By averaging the energy barriers observed in the nine (CS3) and six (CS242) trajectories that exhibited stable ligand-binding under hypersound irradiation with the parameters predominantly used in the simulations (Supplementary Table 2), the activation energies for CS3 and CS242 binding to CDK2 were estimated to be 3.9 ± 1.8 and 6.7 ± 2.4 kcal/mol, respectively ($p = 0.02$, one-sided Student's $t$ test, Table 1), consistent with the relative height of the energy barriers estimated from the free energy landscape (see "Methods"), suggesting that the slower CDK2 association rate of CS242 than CS3 can be attributed to a higher energy barrier. The calculated Arrhenius parameters describing the $k_{on}$ dependence on the temperature indicate that hypersound irradiation increased both the frequency factor (i.e., from $2.2 \times 10^{8}$ to $8.1 \times 10^{8}$ M$^{-1}$ s$^{-1}$ for CS3 and from $2.3 \times 10^{9}$ to $3.6 \times 10^{9}$ M$^{-1}$ s$^{-1}$ for CS242) and the effective temperature (from 298 to 362 K (CS3) and 445 K (CS242), see Table 1). The increase in the frequency factor could be attributed to enhanced ligand diffusion (Table 1)[17], which would increase the collision frequency of the ligand with the protein pocket while enhancing the thermal motions of the solvent molecules did not result in increased ligand diffusion and an acceleration of the ligand-binding process (see "Methods"). In addition, the generation of local high-energy/pressure regions in the solvent leads to an increase in the effective temperature[15]; however, the "macroscopic" temperature of the system remained unchanged at 298 K (Supplementary Table 1). As shown in Supplementary Fig. 4, the native interactions in the CDK2 structure were stably maintained during a series of hypersound-perturbed simulations with different frequencies and amplitudes of shock waves, confirming that the local high-energy regions do not induce thermal denaturation of CDK2. These results suggest that hypersound perturbation accelerates the protein-ligand association process by enhancing the cooperative local motions of the solvent molecules without affecting the native structure of the biomolecules, highlighting the general applicability of this approach for the acceleration of molecular processes in solution.

**Exploration of druggable binding sites on the CDK2 surface.** The identification of previously undiscovered druggable pockets on the protein surface plays a key role in expanding the therapeutic target range of the protein[22]. The above hypersound-perturbed binding simulations allowed us to explore the properties of other sites beyond the ATP-binding pocket on the CDK2 surface. We analyzed the binding simulation data of the ATP-competitive inhibitors CS3 and CS242 and two allosteric inhibitors, 2AN and 9YZ, whose binding sites are distinct from the ATP-binding pocket[23,24]. Hypersound irradiation accelerated the binding of all ligands to both the ATP-binding site and the two allosteric sites 1 and 2 (Fig. 3 and Supplementary Table 4). Allosteric site 2 was frequently accessed by all ligands (Fig. 3A–D), suggesting that this site is remarkably nonspecific because of its shallow pocket shape[24]. In contrast, the CS3/CS242 (Fig. 3A, B) and 2AN (Fig. 3C) ligands showed a more specific preference for the ATP-binding pocket and allosteric site 1, respectively, compared to their nonspecific association with allosteric site 2, supporting the suggestion that these ligands prefer to associate with individual binding sites, as observed in their cocrystal structures[16,23]. The analysis of specific and non-specific sites based on binding simulations of multiple ligands may thus be useful for the prediction of ligand-dependent binding site selectivity and for the exploration of druggable cryptic sites that can allosterically regulate enzymatic activity[22].

**Discussion**
This study shows that hypersound-stimulated MD simulations have the potential to accelerate protein-ligand binding kinetics through a solvent-mediated mechanism without collapse of the protein structure, thus enabling atomic-scale observation of ligand-binding processes within time scales accessible by standard MD (~100 ns). In contrast to other advanced MD methods that accelerate biomolecular processes[25,26], this method does not require prior knowledge of the protein-ligand complex structure. In this way, the simulations can provide significant insights into fundamental biological mechanisms (such as the discovery of microscopic ligand binding pathways involving various bound conformations) and facilitate drug discovery (as illustrated by the present exploration of druggable binding sites on the protein surface). The present acceleration method code is publicly available (see the "Code availability" section), can be implemented on standard high-performance computers, and is suitable for parallel computing because performing multiple

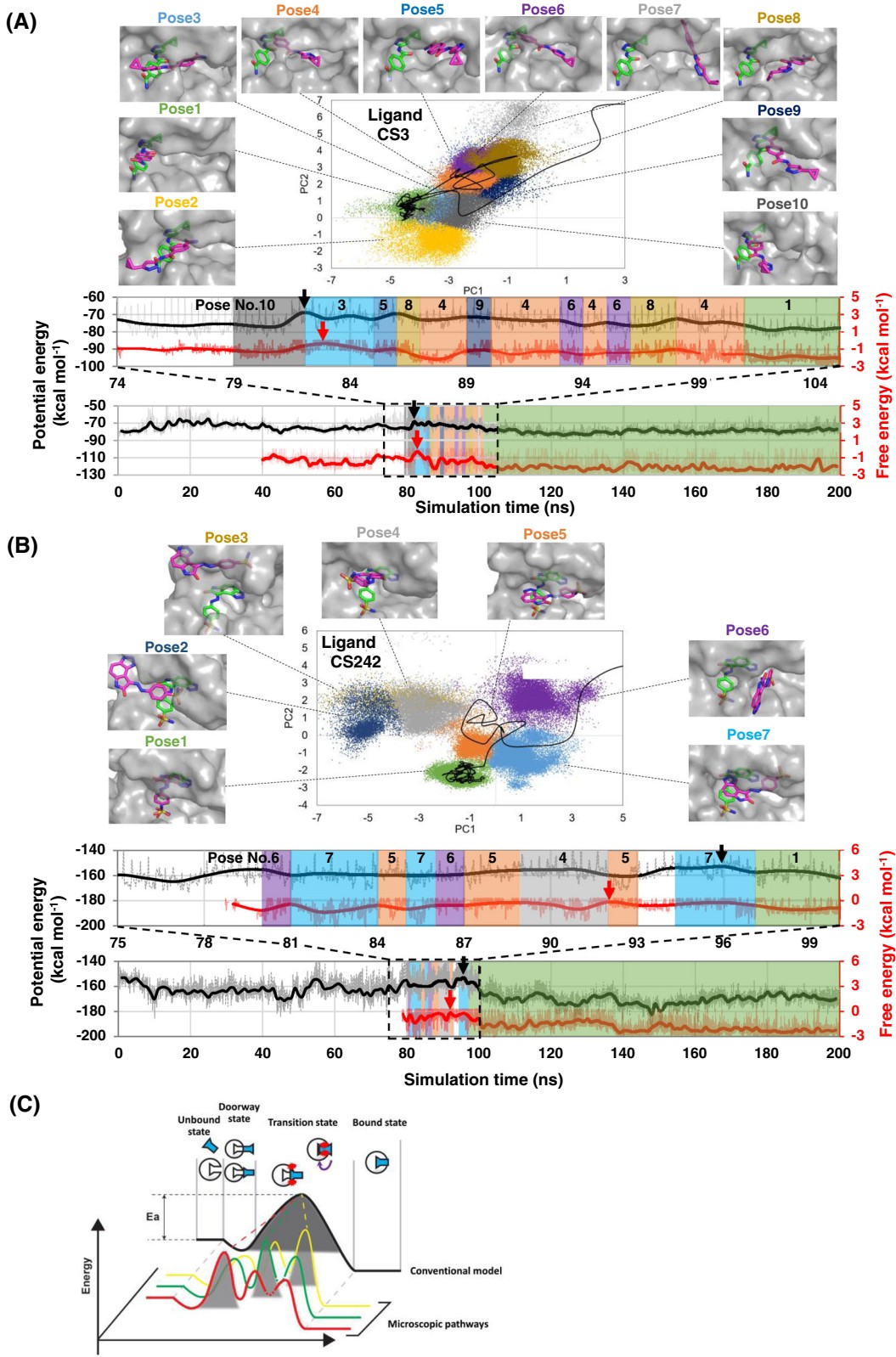

independent simulations in parallel enables the sampling of a higher number of binding events and thus produces an improved statistical description of the process under study. Furthermore, the hypersound irradiation modeled in the simulations is not a fictitious computational procedure, but a real physical process, even though hypersound waves of molecular (several nanometers) wavelength have not yet been realized[17],

which currently hampers the experimental assessment of its impact. Further applications of the present technique to model other biomolecular (e.g., protein conformational changes and protein–protein interactions) and non-biomolecular (e.g., phase transitions of materials) processes are required to assess its general effectiveness in modeling slow dynamic events.

**Fig. 2 Microscopic binding pathways of CDK2 inhibitors. A**, **B** Representative binding pathways of **A** CS3 and **B** CS242 ligands to the ATP-binding pocket of CDK2. (Top) Projections of binding conformations observed in the whole set of MD trajectories (colored dots) and of a representative binding pathway (black line) onto the first and second principal components (PC1 and PC2) calculated from principal component analysis (PCA). Ten (CS3) and 7 (CS242) representative binding poses (magenta sticks) on CDK2 (gray surfaces) are shown alongside the crystallographic pose (green sticks), the closest conformation to which was assigned as Pose 1. (Bottom) Potential energy (black) and free energy (red) trajectories corresponding to the pathway shown in the PCA map. The highest-energy transition state is indicated by a black (potential energy) or red (free energy) arrow. The upper panel shows an enlarged view of these trajectories close to the highest-energy transition state. Note that transition states occur **A** immediately before/after the ligand enters the CDK2 pocket and **B** during conformational rearrangements taking place after pocket entry. **C** Schematic illustration of microscopic and macroscopic kinetic models. The conventional kinetic model assumes a single binding pathway with a single transition state. However, at the single-molecule level, the ligand binds to the protein through multiple pathways with different highest-energy transition state conformations.

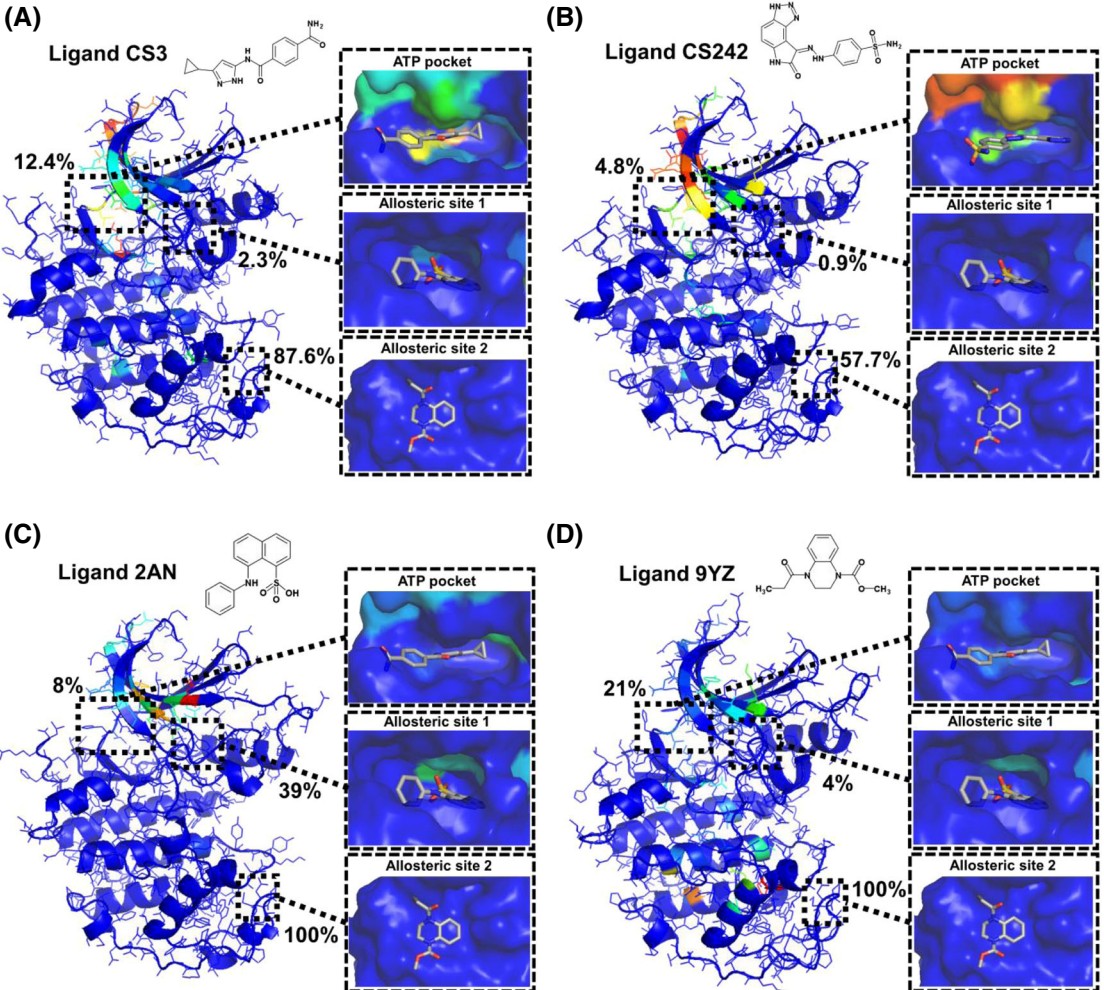

**Fig. 3 Ligand-dependent binding site selectivity. A–D** Specific binding sites of **A** CS3, **B** CS242, **C** 2AN, and **D** 9YZ ligands on the CDK2 surface. The chemical structure of each ligand is shown at the top of the figure. After eliminating residues frequently accessed by all ligands, the backbone (ribbons) and side chains (thin sticks) of CDK2 were colored gradually by the interaction frequency of the individual ligands (e.g., residues with higher and lower frequencies are indicated by red and blue colors, respectively). Enlarged views of the ATP pocket, allosteric site 1, and allosteric site 2 (surfaces) are shown, with the crystallographic poses of CS3/CS242, 2AN, and 9YZ inhibitors (white sticks), respectively. The percentages shown in the models indicate the probabilities of capturing the ligand-binding event in the hypersound-perturbed simulations (Supplementary Table 4).

## Methods

**Model systems and force fields**. We modeled the binding of CDK2 to two ATP-competitive inhibitors, CS3 and CS242, and two allosteric inhibitors, 2AN and 9YZ. The initial structural data of human CDK2 were obtained from the Protein Data Bank (PDB) and the Community Structure-Activity Resource (CSAR) (http://www.csardock.org) databases[16]. Based on cocrystal structures (PDB IDs: 4EK5 (CS3), 4FKQ (CS242), 3PXF (2AN), and 5OO0 (9YZ)), disordered loops and flexible side chains were modeled and refined using the structure preparation module in the MOE program[27], and the dominant protonation state at pH 7.0 was assigned to titratable residues. Considering that a high concentration of ligands enhances the probability of capturing the protein–ligand binding[28], 50 ligands were

randomly placed around the protein and away from the binding site (>17 Å) by translating the ligand in the bound crystal structure.

The ligands were protonated to give net charges of 0 (CS3, CS242, and 9YZ) or −1 (2AN), reflecting the dominant protonation states at neutral pH. GAMESS was used to optimize the structure of each ligand and calculate its electrostatic potential at the HF/6–31G* level[29], after which the atomic partial charges were obtained via the restrained electrostatic potential approach[30]. The other potential parameters of the ligands were obtained by the general AMBER force field[31] using the antechamber module of AMBER Tools 12. The AMBER ff99SB-ILDN force field[32] was used for the protein and ions, while water was modeled with the TIP3P potential[33]. Approximately, 18,000 water molecules were placed around the protein

model in an $8.4 \times 8.4 \times 8.4$ nm$^3$ cubic box. In addition, approximately 60 sodium and chloride ions (corresponding to 150 mM NaCl) were introduced into the simulation box to neutralize all systems, except for the CDK2-2AN complex, for which the NaCl concentration was decreased to 10 mM because of the high concentration of the charged ligand. Based on the volume of the simulation box (592.7 nm$^3$) and the number of ligands (50), the ligand concentration was calculated to be 138 mM, which is much higher than the typical concentrations used in biochemical assays; however, the enhanced ligand diffusion by hypersound irradiation indicates that aggregation of ligand molecules is successfully prevented (Table 1). For the liquid water system, a total of 20,068 water molecules were included in an $8.5 \times 8.5 \times 8.5$ nm$^3$ cubic box.

**Modeling of shock waves.** The isotropic hypersound irradiation of the solute was modeled by generating six different shock waves sequentially propagating from each face of the cubic simulation box ($X_0$, $Y_0$, $Z_0$, $X_1$, $Y_1$, and $Z_1$) toward its center (Fig. 1A, top). Six shock waves were sequentially irradiated in the $+X$, $+Y$, $+Z$, $-X$, $-Y$, and $-Z$ directions, and a delay (corresponding to the $T_{int}$ interval) was applied between each series of shock waves to prevent temperature increase. Each shock wave consisted of five cycles of $16N$ time steps: 80 velocity pulses ($=16 \times 5$ cycles) were applied every $N$ MD step (Fig. 1A, bottom). In each pulse, hypersound-induced velocities are defined as

$$v_i = v_{max} \times \cos(2\pi \times m/16N) \quad (\text{for } i = X_0, Y_0, Z_0, m = 0, N, 2N \dots)$$
$$v_i = -v_{max} \times \cos(2\pi \times m/16N) \quad (\text{for } i = X_1, Y_1, Z_1, m = 0, N, 2N \dots),$$

where $v_{max}$ is the maximum velocity assigned to the pulse and $m$ is the time step number, were added to the thermal velocities of the water molecules located within 1 nm of each surface to model locally originated shock waves. The parameters for shock waves applied to the liquid water and solvated protein–ligand systems are reported in the following subsection and Supplementary Table 2, respectively. A modified version of the GROMACS 4.6.5 program[34] was used to model the shock waves.

**MD simulations.** MD simulations with periodic boundary conditions were carried out using the GROMACS 4.6.5 program on the K computer, Cybermedia Center at Osaka University, and Global Scientific Information and Computing Center at Tokyo Institute of Technology (Japan). Electrostatic interactions were calculated using the particle mesh Ewald method[35] with a cutoff radius of 10 Å, unless stated otherwise; van der Waals interactions were cut off at 10 Å. The P-LINCS algorithm was employed to constrain all bond lengths at their equilibrium value of ref. [36]. After energy minimization, each system was equilibrated as described in the following subsections. A time step of 2 fs was used in all MD runs.

(i)  Liquid water

The system was equilibrated for 1 ns in a constant number of molecules, volume, and temperature (NVT) ensemble. Production runs were also conducted in the NVT ensemble. Electrostatic interactions were cut off at 11 Å. Production runs of 5 ns were performed with and without hypersound irradiation. The $N$, $v_{max}$, and $T_{int}$ parameters in the hypersound-perturbed MD simulations were set to 50, 400 m s$^{-1}$, and 2400$N$, respectively. The cooling effect of Nose–Hoover[37,38], stochastic velocity rescaling[39], and Berendsen[40] thermostats on the hypersound-perturbed MD simulation was examined, showing that all the thermostats with a time constant of 0.1 or 0.3 ps rapidly decreased the excess energy and the elevation of the total kinetic energy returned to the baseline level before the next shock wave pulse (Supplementary Fig. 5). The mass density, pressure, and kinetic energy of the system were analyzed using MD trajectories obtained with a Nose–Hoover thermostat with a time constant of 0.3 ps and calculated using the coordinates and velocities saved every 2 fs.

(ii)  Protein–ligand systems

Each system was equilibrated for 100 ps under NVT conditions, followed by an MD run of 100 ps in a constant number of molecules, pressure, and temperature (NPT) ensemble, with positional restraints applied on protein heavy atoms. Production runs were then conducted under NPT conditions without positional restraints. The temperature was maintained at 298 K by stochastic velocity rescaling[39], and a Parrinello-Rahman barostat was used to maintain the pressure at 1 bar[41]. The temperature and pressure time constants were set to 0.1 and 2 ps, respectively. A total of 283, 369, 100, and 100 independent production runs of 100 ns (with different atomic velocities) were performed for the CDK2-CS3, CDK2-CS242, CDK2-2AN, and CDK2-9YZ systems, respectively. In addition, 1137 (CS3), 362 (CS242), 100 (2AN), and 100 (9YZ) production runs were performed under hypersound irradiation using the parameters summarized in Supplementary Table 2.

**Analysis of MD simulations of liquid water.** The mass density, pressure, and kinetic energy in the hypersound-perturbed MD simulations of liquid water were estimated by focusing on wave propagation along the $X$ direction, as described below.

The mass density was estimated at 82 different $X$-points, based on the number of molecules located within $\pm 0.2$ nm of each point. The kinetic energy ($k_x$) was calculated as $k_x = \frac{1}{2} M \langle v_x \rangle^2$, where $M$ is the mass of a water molecule and $\langle v_x \rangle$ is the $X$ component of the velocity, averaged over all water molecules located within $\pm 0.2$ nm from the corresponding $X$-point. Under hypersound irradiation, $k_x$ was estimated to be 0.4–0.5 kcal/mol at the center of the simulation box ($X = 4$ nm, Fig. 1D). The instantaneous temperature in this region was estimated to be 400–500 K, based on the $k_x$ value of bulk water at 300 K (~0.3 kcal/mol, corresponding to $RT/2$).

The pressure of water in the $+X$ direction of the cubic simulation box was estimated from the $X$ components of the velocities of the water molecules that crossed the $YZ$ plane at a given $X$ during the observation time $\Delta t$, according to the modified van der Waals equation for liquid systems

$$P = \frac{2m}{S\Delta t} \sum_i v_x^i - a \left(\frac{N_A}{V_m}\right)^2 \quad (1)$$

where $m$ is the mass of a water molecule, $S$ is the area of the $YZ$ plane, $v_x^i$ is the $X$ component of the velocity of the $i$th water molecule, $a$ is the intermolecular attractive force constant (described below), $N_A$ is Avogadro's number, and $V_m$ is the molar volume, which was calculated to be 0.0183 L mol$^{-1}$ based on the volume of the simulation box (8.5$^3$ nm$^3$) and the number of water molecules contained in it (20,068). We initially performed a conventional MD simulation of 50 ps, and the first term of Eq. (1), $\frac{2m}{S\Delta t} \sum_i v_x^i$, was calculated to be $1.298 \times 10^8$ Pa based on the water molecules that crossed the $YZ$ plane at $X = 2$ nm (corresponding to the mid-point between the origin and the center of the simulation box) during a $\Delta t$ interval of 50 ps. Using the saturated vapor pressure of water at 298 K ($P = 0.032 \times 10^5$ Pa), the $a$ parameter was estimated to be 0.423 (atm L$^2$ mol$^{-2}$). The pressure under hypersound irradiation was then determined from the hypersound-perturbed MD trajectory, using the estimated value and the sum of the $v_x$ values of the water molecules that crossed the $YZ$ plane at each selected $X$ point during a $\Delta t$ interval of 0.4 ps.

**Analysis of ligand binding within different CDK2 pockets.** For each ligand, we analyzed the MD trajectories of the system containing the CDK2 protein and 50 ligand molecules. Ligand binding within individual CDK2 sites (ATP pocket, allosteric site 1, and allosteric site 2) was considered to occur if at least two distances between an atom belonging to the protein pocket (see below) and any ligand heavy atom were below 5 Å. The following atoms of the protein pocket were used in the distance calculation: Val18 (beta carbon, Cβ) and Leu134 (gamma carbon, Cγ) for the ATP pocket, Tyr15 (zeta carbon, Cζ), and Leu55 (gamma carbon, Cγ) for allosteric site 1, and Cys177 (gamma carbon, Cγ), and Trp227 (indole nitrogen, Nε) for allosteric site 2. Advanced analysis of CS3 and CS242 binding to the ATP pocket is described in the following subsection.

**Advanced analysis of CS3 and CS242 binding to the ATP pocket of CDK2.** For the ATP-competitive inhibitors (CS3 and CS242), whose experimental binding structures and $k_{on}$ values are available, the occurrence of a binding event to the ATP pocket was assessed using stricter criteria, as follows. First, we identified trajectories that satisfied two conditions: (1) a distance between Val18 Cβ and any ligand heavy atom $\leq 5$ Å and (2) the RMSD of the ligand from the crystallographic pose below 9 Å. Next, entry into the ATP pocket was confirmed by visual inspection of these trajectories, using the VMD software[42]. Finally, we identified 67 (CS3) and 14 (CS242) MD trajectories that captured binding events.

In approximately half of these MD trajectories, the bound state was unstable, and the ligand separated from the ATP pocket within 1–40 ns. However, in the remaining trajectories, the ligand remained stably bound to the protein until the end of the simulation; these trajectories were thus extended to 200 ns to further examine the behavior of the bound ligands.

Principal component and conformational clustering analyses of the ligand binding poses observed in representative 27 (CS3) and 14 (CS242) MD trajectories were performed as follows: after removing the overall translation and rotation of the protein, the covariance matrix was calculated using the Cartesian coordinates of the ligand and diagonalized to obtain the PC eigenvectors. The first three principal components (PC1–PC3) accounted for 40%, 33%, and 23% of the variance for CS3, respectively, while PC1–PC3 accounted for 42%, 29%, and 20% of the variance for CS242, respectively. Conformational clustering of the binding poses into an optimal number of clusters was then performed on the first three PCs (PC1–PC3) using the X-means clustering method[43]. The bound states of CS3 and CS242 on the ATP pocket were grouped into 10 and 7 conformational clusters, respectively, one of which corresponded to the crystallographic pose[16], indicating that some of these binding conformations are commonly observed in the 27 (CS3) and 14 (CS242) trajectories.

**Estimation of kinetic parameters for the CDK2–ligand-binding process.** The association rate constant under hypersound irradiation ($k_{on}$), activation energy ($E$), diffusion constant of the solute ($D$), steric factor ($\rho$), frequency factor ($A$), and effective temperature under hypersound irradiation ($T$) were estimated as follows, using the experimental $k_{on}$ values measured without any perturbation and the

trajectories obtained from conventional and hypersound-perturbed MD simulations.

The kinetics of the binding between protein (P) and ligand (L) were analyzed according to the following reaction scheme:

$$P + L \xrightarrow{k_{on}} PL$$

where PL is the protein-ligand complex.

The second-order reaction rate is defined as

$$\frac{d[PL]}{dt} = k_{on}[P][L] \tag{2}$$

where [P], [L], and [PL] are the concentrations of the protein, ligand, and protein–ligand complex, respectively. The initial binding rate is proportional to the initial concentrations of P and L ($[P]_0$ and $[L]_0$, respectively). If $[P] \approx [P]_0$ and $[L] \approx [L]_0$, the following relation can be derived by solving equation [2]

$$\frac{[PL]}{[P]_0} = k_{on}[L]_0 t \tag{3}$$

The $[P]_0$ and $[L]_0$ values in the present simulations of the CDK2–ligand binding were 2.8 and 138 mM, respectively. Based on the experimentally determined $k_{on}$ values of CS3 and CS242 [$3.35 \times 10^5$ and $3.21 \times 10^4$ $M^{-1} s^{-1}$, respectively[16]], the fractions of the CDK2–ligand complexes after 100 ns were expected to be 0.46% (CS3) and 0.044% (CS242). The probabilities of observing the stable ligand binding event in the 100-ns conventional MD simulations of CS3 and CS242 were 0.4% (=1/283) and 0.3% (=1/369) (Supplementary Table 2), respectively. Under hypersound irradiation with $N = 50$ steps, $v_{max} = 400$ m/s, and $T_{int} = 2400N$, which are the parameters predominantly used in the simulations (Supplementary Table 2), and using $[PL]/[P]_0$ ratios of 9/177 (CS3) or 6/227 (CS242), corresponding to the proportions of MD trajectories that exhibited stable ligand binding (Supplementary Table 2), the $k_{on}$ values were estimated to be $3.68 \times 10^6$ (CS3) and $1.92 \times 10^6$ $M^{-1} s^{-1}$ (CS242).

The $k_{on}$ constant can also be described using the Arrhenius equation

$$k_{on} = A\exp{-\frac{E}{RT}} \tag{4}$$

where $R$ is the gas constant. To estimate $E$, the potential energy and free energy differences between the unbound state and the highest-energy transition state were averaged over the 9 (CS3) and 6 (CS242) trajectories. The $E$ values estimated from potential energy trajectories were $3.9 \pm 1.8$ (CS3) and $6.7 \pm 2.4$ (CS242) kcal mol$^{-1}$ ($p = 0.02$, one-sided Student $t$ test), while those estimated from free energy trajectories produced from the free energy landscapes (Supplementary Fig. 6) were $-0.71 \pm 0.23$ (CS3) and $-0.42 \pm 0.18$ (CS242) kcal mol$^{-1}$ ($p = 0.01$, one-sided Student $t$ test). According to a kinetic model involving a "doorway state" located between the unbound and bound states, the frequency factor can be approximated by the diffusion-controlled rate constant[44]

$$A = 4\pi N_A(D_P + D_L)R^*\rho \tag{5}$$

where $N_A$, $D_P$ ($D_L$), $R^*$, and $\rho$ are Avogadro's number, the diffusion constant of the protein (ligand), the critical protein–ligand distance, and the steric factor, respectively. In this study, $R^*$ was set to 1 nm, and we assumed $D_P \ll D_L$. To calculate $D_L$, the mean-square displacement of the 50 ligands during an MD simulation of the solvated CDK2–ligand system was averaged over ten independent simulations. The diffusion constants ($D_{L\_conv}$) of CS3 and CS242 estimated from the conventional MD simulations were $0.17 \pm 0.05 \times 10^{-5}$ and $0.19 \pm 0.07 \times 10^{-5}$ cm$^2$ s$^{-1}$, respectively, while those estimated from the MD runs under hypersound irradiation with $N = 50$ steps, $v_{max} = 400$ m/s, and $T_{int} = 2400$ N ($D_{L\_hyper}$) were $0.61 \pm 0.16 \times 10^{-5}$ (CS3) and $0.30 \pm 0.10 \times 10^{-5}$ cm$^2$ s$^{-1}$ (CS242). Using the $D_{L\_conv}$ and experimental $k_{on}$ values along with the $E$ parameter estimated from the potential energy difference in Eqs. (4) and (5), the steric factors ($\rho$) of CS3 and CS242 were calculated as $10^{-0.76}$ and $10^{0.20}$, respectively. According to Eq. (5), the frequency factors ($A$) without hypersound irradiation were calculated to be $10^{8.35 \pm 0.13}$ $M^{-1} s^{-1}$ (CS3) and $10^{9.36 \pm 0.17}$ $M^{-1} s^{-1}$ (CS242), while those obtained under hypersound irradiation were $10^{8.91 \pm 0.12}$ $M^{-1} s^{-1}$ (CS3) and $10^{9.56 \pm 0.15}$ $M^{-1} s^{-1}$ (CS242). Finally, the effective temperatures under hypersound irradiation calculated from Eq. (4) were 362 K for CS3 and 445 K for CS242.

**Effects of increasing the solvent or solvent/ligand temperature on the probability of observing the ligand-binding event.** To assess how enhancing the thermal motions of the solvent or ligand molecules affects the probability of observing the ligand-binding event, we performed a conventional MD protocol in which the water or ligand diffusion coefficients were adjusted to the values observed in the hypersound-perturbed MD simulation.

The diffusion coefficient of the water molecules in the solvated CDK2-ligand system was estimated to be $4.7 \pm 0.1 \times 10^{-5}$ cm$^2$/s (conventional MDs at 298 K) or $5.5 \pm 0.1 \times 10^{-5}$ cm$^2$/s (hypersound-perturbed MDs with $N = 50$ steps, $v_{max} = 400$ m/s, and $T_{int} = 2400N$). The increased water diffusion coefficient was obtained using a conventional MD protocol in which the temperature of the solvent was increased to 309 K while that of the protein and ligands was maintained at 298 K. This protocol may be the closest to the hypersound-perturbed MD simulation method since additional velocities were only applied to solvent molecules. However,

the diffusion coefficients of the ligands remained $0.18 \pm 0.07 \times 10^{-5}$ cm$^2$/s (CS3) and $0.18 \pm 0.05 \times 10^{-5}$ cm$^2$/s (CS242), which are almost equivalent to those estimated from the conventional MD simulations (i.e., $0.17 \pm 0.05 \times 10^{-5}$ cm$^2$/s for CS3 and $0.19 \pm 0.07 \times 10^{-5}$ cm$^2$/s for CS242 (Table 1)). In addition, the probability of observing the ligand-binding event in this type of conventional MD simulation was estimated to be only 2% (2/100, corresponding to 2 out of 100 MD runs resulting in binding) for CS3 and 3% (3/100) for CS242, which are significantly lower than that in the hypersound-perturbed MD simulations (12.4% for CS3 and 4.8% for CS242 (Supplementary Table 2)).

The diffusion coefficients of CS3 and CS242 estimated from the MD runs under hypersound irradiation with the same parameters described above were $0.61 \pm 0.16 \times 10^{-5}$ cm$^2$/s (CS3) and $0.30 \pm 0.10 \times 10^{-5}$ cm$^2$/s (CS242) (Table 1). The diffusion coefficients close to these values were obtained using a conventional MD protocol in which the temperature of the ligands and solvent was increased to 375 K (CS3) or 355 K (CS242), while that of the protein was maintained at 298 K. The probability of observing the ligand-binding event in this type of conventional MD simulation was estimated to be 14% (14/100) for CS3 and 4% (4/100) for CS242, which is almost equivalent to that in the hypersound-perturbed MD simulations. However, this type of simulation (different temperatures of protein and ligand/solvent) is unrealistic, and also induces a partial collapse of rigidly structured regions in CDK2 (Supplementary Fig. 7), because of the excessively increased diffusion coefficient of water molecules ($11.6 \pm 0.1 \times 10^{-5}$ cm$^2$/s at 375 K or $9.5 \pm 0.1 \times 10^{-5}$ cm$^2$/s at 355 K). On the other hand, when the ligands and solvent were coupled separately to temperature baths at 375 K/355 K and 309 K, respectively, the diffusion coefficients of the ligands remained $0.21 \pm 0.09 \times 10^{-5}$ cm$^2$/s (CS3) and $0.22 \pm 0.05 \times 10^{-5}$ cm$^2$/s (CS242). This is presumably because stably formed hydrogen bond networks in water near room temperature would hamper free diffusion of ligand molecules even if their kinetic energy is enhanced. Therefore, drastically increasing the thermal motions of solvent molecules (i.e., 375/355 K) appears to be required for significant enhancement of ligand diffusion, demonstrating distinct effects from hypersound irradiation, which induces a significant acceleration in ligand diffusion by moderately enhancing cooperative local motions of solvent molecules.

**Identification of specific ligand binding sites on the CDK2 surface.** The specific binding sites of each of the ATP-competitive inhibitors (CS3 and CS242) and allosteric inhibitors (2AN and 9YZ) on the CDK2 surface were determined as follows. First, the root-mean-square fluctuation of the ligand was calculated every 10 ns of the individual 100-ns hypersound-perturbed MD trajectories obtained with $N = 50$ steps, $T_{int} = 2400N$, and $v_{max} = 400$ m/s (CS3, CS242, and 9YZ) or $v_{max} = 300$ m/s (2AN). If the value was below 3 Å, a stable CDK2–ligand complex was considered to be formed during the 10-ns period, and residues that interact with the ligand (<5 Å) were extracted from the mean coordinates of the protein and ligand. Next, the frequency of ligand interactions at each CDK2 residue ($f_{int}$) was calculated across all stable complex structures and normalized by the number of MD trajectories. Finally, after excluding residues that frequently interacted with all ligands ($f_{int}$ of more than 0.1) as nonspecific binding sites, residues with higher $f_{int}$ values were identified as specific binding sites.

**Reporting summary**. Further information on research design is available in the Nature Research Reporting Summary linked to this article.

## Data availability
Data supporting the findings of this manuscript are available from the corresponding authors upon reasonable request. A reporting summary for this Article is available as a Supplementary Information file. Source data are provided with this paper. The initial structural data of human CDK2 are publicly available in the Protein Data Bank (PDB) (https://www.rcsb.org/) or the Community Structure-Activity Resource (CSAR) (http://www.csardock.org) databases. Molecular dynamics data (the input files, MD trajectories, and processed data) are available in the Biological Structure Model Archive under BSM-00027 (https://bsma.pdbj.org/entry/27) or our laboratory server at https://bmdi-db.med.kyoto-u.ac.jp/owncloud/index.php/s/L8rwegnll6yXj5l.

## Code availability
The hypersound-perturbed MD code is available free of charge at https://github.com/clinfo/gromacs (https://doi.org/10.5281/zenodo.4646306).

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

## Acknowledgements

We thank J. Higo and I. Fukuda for the critical reading of the paper. This study was supported by the Ministry of Education, Culture, Sports, Science and Technology (MEXT, Japan) projects "Priority Issue on Post-K Computer (Building Innovative Drug Discovery Infrastructure through Functional Control of Biomolecular Systems)" and "Program for Promoting Researches on the Supercomputer Fugaku (Application of Molecular Dynamics Simulation to Precision Medicine Using Big Data Integration System for Drug Discovery)" (to Y.O.), Foundation for Computational Science (FOCUS) Establishing Supercomputing Center of Excellence (to Y.O.), the K supercomputer-based drug discovery project by Biogrid pharma consortium (to Y.O.), and a Japan Society for the Promotion of Science (JSPS) KAKENHI Grant (Nos. JP18K06594 and JP21K06510) (to M.A). The simulations were carried out on the K computer and HPCI systems provided by the RIKEN, Osaka University (VCC), and Tokyo Institute of Technology (TSUBAME), through the HPCI System Research Project (project IDs: hp140042, hp150025, hp150272, hp160213, hp170275, hp180186, hp190154, hp200011, hp200129, and ra000018).

## Author contributions

M.A. designed and performed the simulations. M.A., S.M., G.B., Y.I., Y.S., and N.K. analyzed the simulation data. M.A. wrote the paper. M.A. and Y.O. supervised the study. All the authors discussed the research, edited the paper, and approved its final version.

## Competing interests

The authors declare no competing interests.
