## [Peer Review File · Nature Communications]

Reviewer #1 (Remarks to the Author):

The authors have addressed my concerns and the manuscript is improved.

However, the following points require attention:

-The authors have done useful tests of the effects of changing the diffusion D of the water and the ligands by increasing the temperatures of the respective temperature baths. They note that when the ligand and solvent are separately put to temperatures corresponding to their diffusion rates in hypersound-accelerated MD and then conventional MD is run, the ligands have a much lower diffusion rate but no explanation is given. Can the authors explain why this is so given that the ligands are maintained with a certain kinetic energy in their own temperature bath? Is this related to the artificially high ligand concentration used? If so, conventional MD could be performed with fewer ligands or with removal of interactions between ligands or additional non-specific repulsion between ligands to prevent aggregation. It is not clear why this is not as effective, from a computational efficiency perspective, as hypersound-accelerated molecular dynamics (which still requires a significant total simulation time (hundreds of 100 ns trajectories totaling 10s of microseconds per ligand for the optimal parameters (supplementary table 2))).

-Table 1: The use of ' $\log(P)$ ' for the steric factor is confusing as $\log P$ is commonly used for the octanol-water partition coefficient of compounds. Also P is used to represent protein, pressure and probability density in the Supplementary information. A consistent and unambiguous set of symbols should be used throughout the manuscript.

-Table 1: As far as I understand, the activation energy was determined from the hypersound-accelerated MD simulations, not the conventional MD simulations. Therefore a footnote is required to make clear the assumption that the same activation energy applies to the conventional MD.

-To avoid confusion with the 'accelerated molecular dynamics' method developed in the McCammon group, all mentions of 'accelerated molecular dynamics' should be changed to 'hypersound-accelerated molecular dynamics'

-p7: 'without a significant decrease in computation speed': this should be 'increase'

Reviewer #2 (Remarks to the Author):

The authors have done an excellent job at convincingly addressing all points that I raised with the previous submission. I have no further concerns and congratulate the authors to this exciting contribution.

Reviewers' comments to Author & Reply to reviewers

To Reviewer #1

Thank you very much for your additional comments on our revised manuscript.

Remarks to the Author:

The authors have addressed my concerns and the manuscript is improved.

However, the following points require attention:

Comment (1) The authors have done useful tests of the effects of changing the diffusion D of the water and the ligands by increasing the temperatures of the respective temperature baths. They note that when the ligand and solvent are separately put to temperatures corresponding to their diffusion rates in hypersound-accelerated MD and then conventional MD is run, the ligands have a much lower diffusion rate but no explanation is given. Can the authors explain why this is so given that the ligands are maintained with a certain kinetic energy in their own temperature bath? Is this related to the artificially high ligand concentration used? If so, conventional MD could be performed with fewer ligands or with removal of interactions between ligands or additional non-specific repulsion between ligands to prevent aggregation. It is not clear why this is not as effective, from a computational efficiency perspective, as hypersound-accelerated molecular dynamics (which still requires a significant total simulation time (hundreds of 100 ns trajectories totaling 10s of microseconds per ligand for the optimal parameters (supplementary table 2)).

Answer: Results of high temperature MD simulations performed in response to the previous reviewer's concern have been described in the third paragraph under the "Effects of increasing solvent or solvent/ligand temperature on the probability of observing ligand binding events" subheading within the Methods section. These high temperature MD simulations showed that increasing ligand and solvent temperatures up to 375 K (CS3) or 355 K (CS242) enhanced ligand diffusion even at high ligand concentrations (diffusion coefficients obtained were $0.59 \pm 0.19 \times 10^{-5} \text{ cm}^2/\text{s}$ (CS3) and $0.28 \pm 0.16 \times 10^{-5} \text{ cm}^2/\text{s}$ (CS242), close to those estimated from the hypersound-perturbed MDs), while increasing the temperature of ligands alone did not

(diffusion coefficients were $0.21 \pm 0.09 \times 10^{-5} \text{ cm}^2/\text{s}$ (CS3) and $0.22 \pm 0.05 \times 10^{-5} \text{ cm}^2/\text{s}$ (CS242) for ligands at 375K/355K and solvent at 309K). These results suggest that heating both ligands and solvent to the same high temperature (375K/355K) is required for significant enhancement of ligand diffusion. This is presumably because stably formed hydrogen bond networks in water near room temperature would hamper free diffusion of ligand molecules even if their kinetic energy is enhanced, and thus drastically increasing the thermal motions of solvent molecules (i.e. 375K/355K), which results in the excessively increased diffusion coefficient of them ($11.6 \pm 0.1 \times 10^{-5} \text{ cm}^2/\text{s}$ at 375 K or $9.5 \pm 0.1 \times 10^{-5} \text{ cm}^2/\text{s}$ at 355 K), appears to be required for significant enhancement of ligand diffusion. By contrast, hypersound-perturbed MD simulations characteristically induced a significant acceleration in ligand diffusion by moderately enhancing cooperative local motions of solvent molecules, improving proteins' efficiency at capturing ligands by 10–20 fold compared with standard MD simulations, although tens of microseconds of simulation are still required to achieve a protein-ligand pair.

One reason why heating both ligands and solvent to the same high temperature in the conventional MD protocol is required for significant enhancement of ligand diffusion has been included in the “Effects of increasing solvent or solvent/ligand temperature on the probability of observing ligand binding events” subheading within the Methods section.

Comment (2) Table 1: The use of 'log(P)' for the steric factor is confusing as logP is commonly used for the octanol-water partition coefficient of compounds. Also P is used to represent protein, pressure and probability density in the Supplementary information. A consistent and unambiguous set of symbols should be used throughout the manuscript.

Answer: In response to the reviewer's comments, the symbol of the steric factor (P) has been changed to “ p ”, which is generally used to represent this factor.

Comment (3) Table 1: As far as I understand, the activation energy was determined from the hypersound-accelerated MD simulations, not the conventional MD simulations. Therefore a footnote is required to make clear the assumption that the same activation energy applies to the conventional MD.

Answer: In response to the reviewer's comments, we have included a footnote below Table 1 that clearly shows how we estimated activation energies, and states our assumption that activation energy is not affected by hypersound irradiation.

Comment (4) To avoid confusion with the 'accelerated molecular dynamics' method developed in the McCammon group, all mentions of 'accelerated molecular dynamics' should be changed to 'hypersound-accelerated molecular dynamics'.

Answer: All mentions of “accelerated molecular dynamics” have been replaced with “hypersound-accelerated molecular dynamics”.

Comment (5) 'without a significant decrease in computation speed': this should be 'increase'.

Answer: Since computation speed for hypersound-perturbed MDs is slightly slower than that for standard MDs (Supplementary Table 3), the statement “without a significant decrease in computation speed” has been replaced with “without significant loss of computation speed”.

To Reviewer #2

Remarks to the Author:

The authors have done an excellent job at convincingly addressing all points that I raised with the previous submission. I have no further concerns and congratulate the authors to this exciting contribution.